# Cholinergic neuromodulation of inhibitory interneurons facilitates functional integration in whole-brain models

**Carlos Coronel-Oliveros**[1,2], **Rodrigo Cofré**[3], **Patricio Orio**[1,4]*

1 Centro Interdisciplinario de Neurociencia de Valparaíso, Universidad de Valparaíso, Valparaíso, Chile, 2 Programa de Doctorado en Ciencias, mención Biofísica y Biología Computacional, Universidad de Valparaíso, Valparaíso, Chile, 3 CIMFAV-Ingemat, Facultad de Ingeniería, Universidad de Valparaíso, Valparaíso, Chile, 4 Instituto de Neurociencias, Facultad de Ciencias, Universidad de Valparaíso, Valparaíso, Chile

* patricio.orio@uv.cl

**Data Availability Statement:** All relevant data are within the manuscript and its Supporting information files. The code we use to obtain our

## Abstract

Segregation and integration are two fundamental principles of brain structural and functional organization. Neuroimaging studies have shown that the brain transits between different functionally segregated and integrated states, and neuromodulatory systems have been proposed as key to facilitate these transitions. Although whole-brain computational models have reproduced this neuromodulatory effect, the role of local inhibitory circuits and their cholinergic modulation has not been studied. In this article, we consider a Jansen & Rit whole-brain model in a network interconnected using a human connectome, and study the influence of the cholinergic and noradrenergic neuromodulatory systems on the segregation/integration balance. In our model, we introduce a local inhibitory feedback as a plausible biophysical mechanism that enables the integration of whole-brain activity, and that interacts with the other neuromodulatory influences to facilitate the transition between different functional segregation/integration regimes in the brain.

## Author summary

Segregation of brain activity refers to the fact that some brain regions are specialized to handle particular features of external and internal stimuli. However, to produce a coherent behavioral outcome, the brain must coordinate the activity of these specialized brain areas, and this is called integration of brain activity. Based on a fixed connectome (the brain anatomical structure), the neuromodulatory systems are one of the plausible candidates to manage the transitions of brain states in short timescales. Understanding the role of neuromodulators in brain dynamics and the segregation/integration balance is relevant, in particular, as it is known that in several neuropsychiatric disorders the segregation/integration balance its impaired. Here, we used a computational model of the whole brain to study the dual effect of the cholinergic and noradrenergic neuromodulatory systems in the switching from segregated to integrated brain states. The novelty of our work is the inclusion of a homeostatic local inhibitory loop. This specific inhibition, modulated

results is open source and available at https://github.com/vandal-uv/Neuromod2020.

**Funding:** This work was supported by Fondo Nacional de Desarrollo Científico y Tecnológico - Fondecyt Grants 1181076 (to PO) and 11181072 (to RC) and the Advanced Center for Electrical and Electronic Engineering - ANID (FB0008 to PO). The Centro Interdisciplinario de Neurociencia de Valparaíso (CINV) is a Millenium Institute supported by the ANID grant ICN09_022. CC-O is funded by Beca Doctorado Nacional – ANID grant 2018- 21180995. The funders had no role in study design, data collection and analysis, decision to publish, or preparation of the manuscript.

**Competing interests:** The authors have declared that no competing interests exist.

by the cholinergic system, maintains the excitation/inhibition balance while promoting integration. Our work links the local effects of cholinergic neuromodulation, with the more global influences of the structural connectivity and neuromodulatory systems. This constitutes a step forward in the understanding of the neural mechanisms behind the segregation/integration balance of brain activity.

## Introduction

Integration and segregation of brain activity are nowadays two well-established brain organization principles [1–4]. Functional segregation refers to the existence of specialized brain regions, allowing the local processing of information. Integration coordinates these local activities in order to produce a coherent response to complex tasks or environmental contexts [1, 2]. Both segregation and integration are required for the coherent global functioning of the brain; the balance between them constitutes a key element for cognitive flexibility, as highlighted by the theory of coordination dynamics [5, 6].

From a structural point of view, the complex functional organization of the brain is possible thanks to an anatomical connectivity that combines both integrated and segregated network characteristics, having small-world and modular properties [7]. In spite of this *structural connectivity* (SC) remaining fixed over short timescales, different patterns of *functional connectivity* (FC) can be observed during the execution of particular behavioral tasks [2]. Moreover, functional Magnetic Resonance Imaging (fMRI) neuroimaging studies show that during a resting state the FC is not static, but rather evolves over the recording time [8–10], highlighting the non-linear and non-stationary properties of the FC [11]. In a similar way, the integration and segregation of brain activity are not static over time [3, 12]. In this context, an interesting question emerges: *How does the brain manage to produce dynamical transitions between different functional states from a rigid anatomical structure?*.

Neuromodulatory systems tune the firing properties of neurons, providing a mechanism to change the flow of information within the brain, and allowing the transitions between different FC patterns. A recent hypothesis proposed by Shine [13] argues that neuromodulation allows the transition between integrated and segregated states, manipulating the neural gain function [14]. In that line, the cholinergic and noradrenergic systems have been proposed as candidates to influence the cognitive processing within the brain [15, 16], in spite of not being the unique neuromodulatory systems in the central nervous system which can tune the firing properties of neurons [14, 17].

The cholinergic system is involved in cognitive and attentional selectivity [16], and in the cerebral cortex the main source of acetylcholine are projections from the basal forebrain [18]. Acetylcholine increments the overall excitability [19, 20], and consequently rises population activity above noise, a mechanism referred as response gain [14]. The increase in signal-to-noise ratio, especially in brain areas that are close to each other, promotes segregation when considering the response gain by itself [13]. On the other hand, the noradrenergic system is related to the exploratory behavior [15], and the principal source of noradrenergic projections to the cerebral cortex comes from the locus coeruleus [21]. Noradrenaline increases the reponsivity (or selectivity) of neuronal populations to input-driven activity (e.g., sensory stimuli, inputs for distant brain areas relevant to a task) with respect to spontaneous activity (or the internal state of the brain) [22–24], filtering out noise [25] in a mechanism called filter gain [14]. The effect of the noradrenergic system in increasing the signal-to-noise ratio facilitates the detection of signals embedded in a noisy environment [25], boosting the signal detection

and promoting integration [13]. Therefore, a complex interaction between the cholinergic and noradrenergic system seems to manage the balance between integration and segregation.

Using a whole-brain model, Shine *et al.* [26] showed that neuromodulation and integration follow an inverted-U relationship. If one considers that neuromodulation also shows an inverted-U relationship with in-task performance, [15, 27], it is possible to hypothesize that neuromodulatory systems boost cognitive and attentional performance by increasing the functional integration in the brain, as proposed by Shine *et al.*. [13, 26].

There are still unanswered questions about the specific effects of neuromodulation on integration and segregation. Experimental research points out that the cholinergic system, through both nicotinic and muscarinic receptors, boosts the signal-to-noise ratio in two principal ways [14, 28]: first, increasing the excitability of pyramidal neurons [29–31], and second, enhancing the firing rates of dendritic-targeting GABAergic interneurons –an effect that promotes a focused intra-columnar inhibition, reducing the local excitatory feedback to pyramidal neurons [31–33]. Consequently, pyramidal neurons become more responsive to stimulus from other distant regions respect to the stimulus of its own cortical column [28, 29, 34]. The particular effect of the cholinergic system on excitatory neurons was one of the focus of the whole-brain simulation work by Shine *et al.*. [26]. However, the cholinergic modulation of inhibitory interneurons and its effect on the segregation/integration balance has not been analyzed at the whole-brain level. This is the main focus of the present work.

Here, we use an *in silico* approach to analyze the effect of neuromodulatory systems on functional integration in the brain, focusing on the cholinergic action in inhibitory interneurons. We combined a real human structural connectivity with the Jansen & Rit neural mass model of cortical columns [35, 36]. The mesoscopic properties of the model enable us to study more specifically the effects of neuromodulators in whole-brain dynamics. To make our simulations more comparable to experimental findings [3, 12, 37, 38], and also following Shine *et al.* [26], fMRI blood-oxygen-level-dependent (BOLD) signals were generated from the firing rates of pyramidal neurons. Integration and segregation were then assessed in the functional connectivity matrices derived from the BOLD-like signals, using a graph theoretical approach.

The neuromodulation was discerned in three components. First, we included an "excitatory gain", which increases the inter-columnar coupling. In our model, this gain mechanism is mediated by the action of the cholinergic system on pyramidal neurons, having an indirect effect on their excitability [13, 14, 28]. Second, we added an "inhibitory gain", also mediated by the cholinergic system, that controls the inputs from inhibitory to excitatory interneurons and reduces the local feedback excitation. This additional connection, well described in cortical columns [39, 40], represents a modification of the original neural mass model proposed by Jansen & Rit [35, 36]. Finally, we incorporated a "filter gain", that increments the pyramidal neurons' sigmoid function slope [14]. This gain mechanism is mediated by the noradrenergic system; it acts as a filter, decreasing (increasing) the responsivity to weak (strong) stimuli [23, 25], boosting signal-to-noise ratio. We show that the increase of the signal-to noise ratio, mediated both by the excitatory and filter gains, and the decrease in the feedback excitation, related to the inhibitory gain, promote functional integration.

## Results

We assessed the effect of the neuromodulatory systems using a whole-brain neural mass model of brain activity. In the model, each node corresponds to a brain area and is represented by a neural mass consisting of three populations [35, 36]: pyramidal neurons (that reside in cortical column layer V), excitatory interneurons (nearby pyramidal cells which reside in the same layer than the principal pyramidal population), and inhibitory interneurons (Fig 1A).

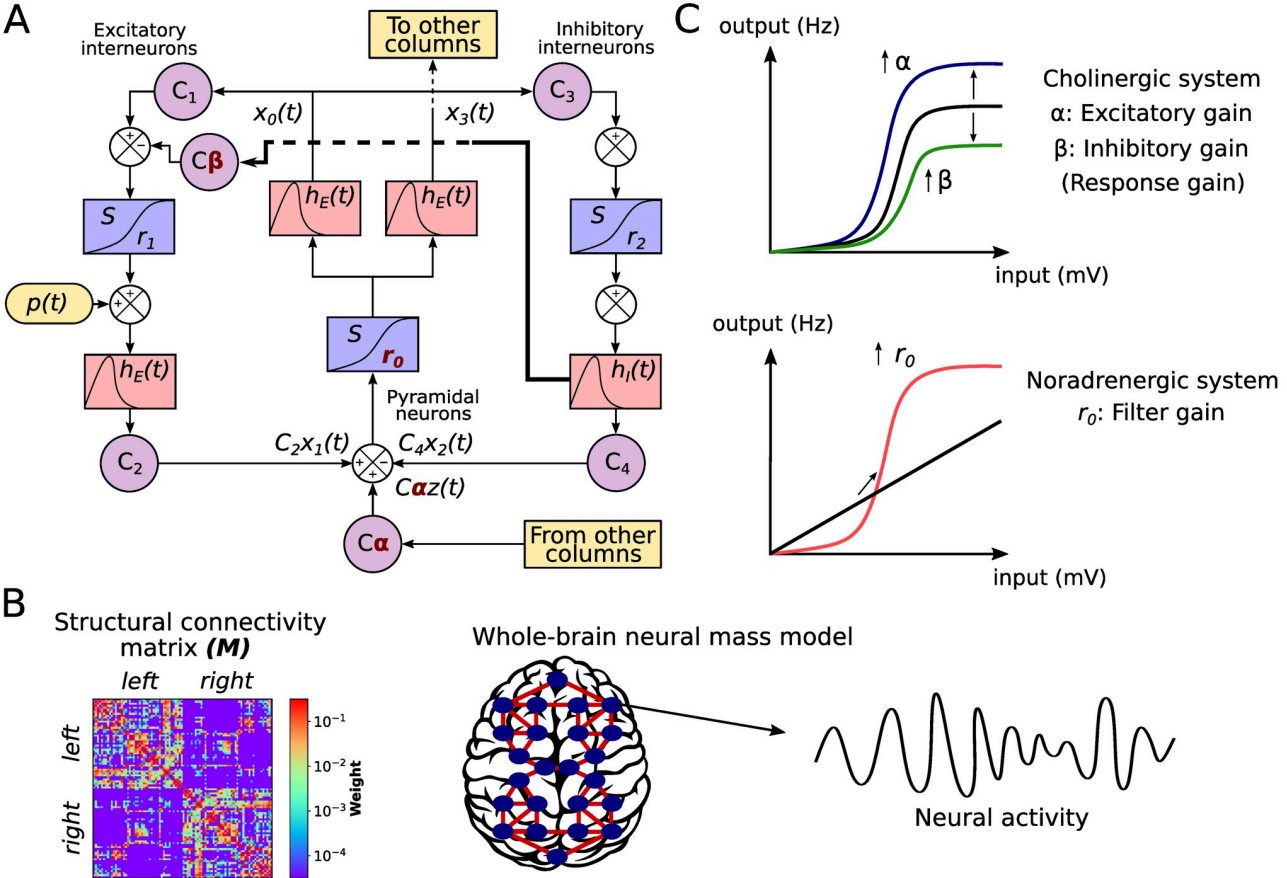

**Fig 1. Whole-brain neural mass model. A)** The Jansen & Rit model is constituted by a population of pyramidal neurons with excitatory and inhibitory feedback mediated by interneurons (INs). Each population is connected by a series of constants $C_i$. The outputs are transformed from average pulse density to average postsynaptic membrane potential by an excitatory (inhibitory) impulse response function $h_E(t)$ ($h_I(t)$). Then, a sigmoid function $S$ performs the inverse operation. Pyramidal neurons project to distant cortical columns, and receive both uncorrelated Gaussian-distributed inputs $p(t)$ and inputs from other cortical columns $z(t)$. Neuromodulation is constituted by three parameters, colored in red: excitatory gain $\alpha$, which scales $z(t)$, inhibitory gain $\beta$, which increases the inhibitory input to excitatory INs (thick line), and filter gain, $r_0$, which modifies the slope of the sigmoid function in pyramidal neurons. **B)** Each node represents a cortical column, whose dynamics is ruled by the Jansen & Rit equations. Nodes are connected through a structural connectivity matrix, $M$ **C)** Neuromodulation modifies the coupling between neurons and the properties of the input (average postsynaptic membrane potential) to output (average pulse density) sigmoid function. The cholinergic system modifies the global coupling and local inhibition. $\alpha$ amplifies the response of pyramidal neurons to other columns' input; it also increases pyramidal neurons excitability. $\beta$ amplifies the effect of inhibitory INs to excitatory INs, damping pyramidal cells excitability. The noradrenergic system increments the responsivity of pyramidal neurons to relevant stimuli respect to noise, as a filter, by increasing the slope $r_0$ of their sigmoid function.

Based on Silberberg & Markram [39] and Fino *et al.* [40], we have added a connection from inhibitory interneurons from excitatory interneurons (thick line in Fig 1A), allowing us to study the effect of its modulation by cholinergic influences (see below). The nodes are connected through a weighted and undirected structural connectivity matrix derived from human data [41], parcellated in 90 cortical and sub-cortical regions with the automated anatomical labeling (AAL) atlas [42] (Fig 1B). Connections between nodes are made by pyramidal neurons, considering that long-range projections are mainly excitatory [43, 44]. Using the firing rates of each node as inputs to a generalized hemodynamic model [45], we obtained fMRI BOLD signals from which we calculated integration and segregation of the resulting FC matrices.

Following Shine *et al.* [26], we modeled the influence of the cholinergic and noradrenergic systems through the manipulation of the response and filter gains, respectively (Fig 1C). The

principal difference in our approach is that we split the response gain in excitatory gain (long-range pyramidal to pyramidal coupling), $\alpha$, and inhibitory gain (local inhibitory to excitatory interneurons coupling), $\beta$. While the excitatory gain boosts the pyramidal neurons responsivity to long-range inputs, and indirectly increases the pyramidal cells excitability, the inhibitory gain reduces the local excitatory feedback from interneurons. Finally, the filter gain $r_0$ modifies the sigmoid function slope of pyramidal neurons, increasing its responsivity to relevant stimuli and boosting signal-to-noise ratio. Here, we studied the combined effect of the three gain mechanisms to understand how neuromodulatory systems shape the global neuronal dynamics in two different timescales: EEG-like and BOLD-like signals. Our hypothesis is that the inhibitory gain will play a significant role in increasing the likelihood of integration.

## Inhibitory gain facilitates neuronal coordination

We first studied the combined influence of the excitatory and inhibitory response gains, by fixing $r_0 = 0.56$ mV$^{-1}$ (its default value) and then simulating neuronal activity at different combinations of $\alpha \in [0, 1]$ and $\beta \in [0, 0.5]$. Then, we analyzed the graph properties of the *static* (time-averaged) functional connectivity (sFC) matrices, obtained from the pairwise Pearson's correlations of BOLD-like signals. Namely, we calculated the global efficiency $E^w$, a measure of integration defined as the inverse of the characteristic path length [46], and modularity $Q^w$, a measure of segregation based on the detection of network communities or modules [46]. High values of $E^w$ represent an efficient coordination between all pairs of nodes in the network, a signature of integration. In contrast, a high modularity $Q^w$ is associated to segregation [46].

Fig 2A shows that functional integration ($E^w$) is maximized in an intermediate region of the ($\alpha, \beta$) parameter space; and that integration is accompanied by a decrease in the segregation ($Q^w$). Also, the system undergoes a sharp transition crossing a critical boundary. The transitions between different regimes are better appreciated in Fig 2B, where we show a 1-D sweep of $\alpha$ at $\beta = 0.25$. Dashed lines at $\alpha = 0.3$ and $\alpha = 0.8$ correspond to points in the parameter

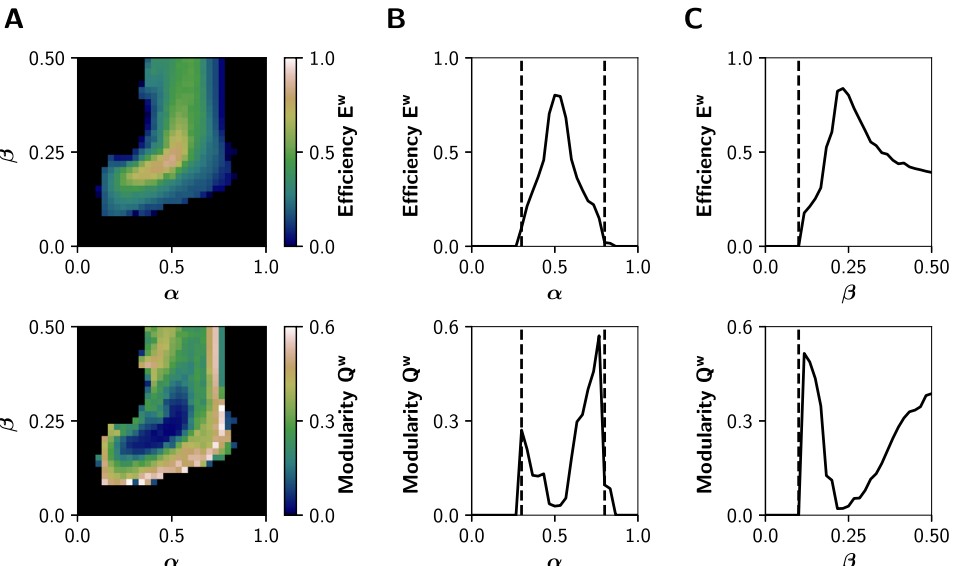

**Fig 2. Network features in the ($\alpha, \beta$) parameter space. A)** Global efficiency $E^w$ (integration) and modularity $Q^w$ (segregation) of the graphs derived from the sFCs of the BOLD-like signals. **B)** Transitions through critical boundaries in the $\alpha$ axis, for a fixed $\beta = 0.25$. Transition points are represented by black dashed lines at $\alpha = 0.3$ and $\alpha = 0.8$. **C)** Transitions in the $\beta$ axis, for a fixed $\alpha = 0.5$, with a critical transition at $\beta = 0.1$.

space where drastic changes in dynamic properties of the network occur. Further, the global efficiency $E^w$ follows an inverted-U relationship with the excitatory neuromodulation, suggesting (in our whole-brain model) that optimal levels of cholinergic neuromodulation maximize functional integration (see Discussion). Also, $Q^w$ peaks higher at the right critical boundary (dashed lines). A 1-D sweep of $\beta$ at $\alpha = 0.5$ (Fig 2C), shows an increase in integration crossing the critical transition at $\beta = 0.1$. These results are replicated using other measures of integration and segregation: the mean participation coefficient $PC^w$, an integration metric that quantifies the between-modules connectivity, and the transitivity $T^w$, which accounts for segregation counting triangular motifs [46] (S1 Fig).

The modulation of the inhibitory gain ($\beta$) shows an important effect on the integration and segregation properties of the whole network measured by the global efficiency and modularity, respectively. This could be due to the reduction of excitatory feedback only, or to a more specific effect of the newly introduced connection from inhibitory to excitatory interneurons. In the first case, we expect a similar effect by reducing the $C_1$ parameter (see Fig 1A) because this also reduces the excitatory feedback loop of the cortical columns. As shown in S2 Fig, this is only partially the case. The reduction of the $C_1$ connection weight –in the absence of the inhibitory to excitatory interneuron connection– enables the network to reach integration but in a smaller region of the parameter space and to a lower extent than the inhibitory modulation that we introduced in our model.

To show in more detail how each gain mechanism produces integrated or segregated functional network states, we present in Fig 3 some BOLD-like signals and their respective sFC matrices. We chose five tuples of ($\alpha$, $\beta$) parameters, marked with the red circles in the Fig 3A. Functional integration measured by global efficiency is maximal in the middle ($\alpha = 0.5$, $\beta = 0.25$), and segregation measured by modularity is maximized far away from this point ($\alpha = 0.25$, $\beta = 0.125$, and $\alpha = 0.75$, $\beta = 0.375$). In the extreme cases ($\alpha = 0$, $\beta = 0$, and $\alpha = 1$, $\beta = 0.5$) there is neither integration nor segregation; in the first case the network is disconnected, and in the second one the system crossed the second bifurcation point and pyramidal neurons are not oscillating (neurons are over-excited).

## Inhibitory gain allows the noradrenaline-mediated integration

To further validate our model, we sought to reproduce the results of the neuromodulatory paradigm proposed by Shine *et al.*. [13, 26]. We characterized the relationship between neuromodulation and integration in the ($\alpha$, $r_0$) parameter space, with $\alpha \in [0, 1]$ and $r_0 \in [0, 1]$ while leaving $\beta$ fixed at 0 or 0.4 (without and with inhibitory gain, respectively). The results for $\beta = 0$ (Fig 4A) show no integration in the entire parameter space. On the other hand, the observations of Shine *et al.* [26] are fully reproduced with $\beta = 0.4$ (Fig 4B). Similar results hold for the mean $PC^w$ and $T^w$, as shown in S3 Fig.

As observed previously in Fig 2, critical boundaries delimit asynchronous and synchronous states in the ($\alpha$, $r_0$) parameter space. A 1-D sweep of $\alpha$ at $r_0 = 1$ mV$^{-1}$ shows a sharp transition (Fig 4C) (Fig 4B shows that this is also true for lower values of $r_0$). Global efficiency $E^w$ increases alongside the decrease of modularity $Q^w$, and further increments of $\alpha$ produce network desynchronization. On the other hand, a 1-D sweep of $r_0$ at $\alpha = 0.6$ (Fig 4D) produces similar observations, but just one boundary is visible. As in the ($\alpha$, $\beta$) parameter space, the excitatory gain $\alpha$ follows an inverted-U relationship with integration. This relationship was not observed between the filter gain $r_0$ and integration.

In the whole-brain model, the cholinergic system exerts its effect by changing both $\alpha$ and $\beta$ parameters. Under this assumption, a logical consequence of the cholinergic neuromodulation is the possibility of $\alpha$ and $\beta$ increasing/decreasing simultaneously. For that reason, we repeated

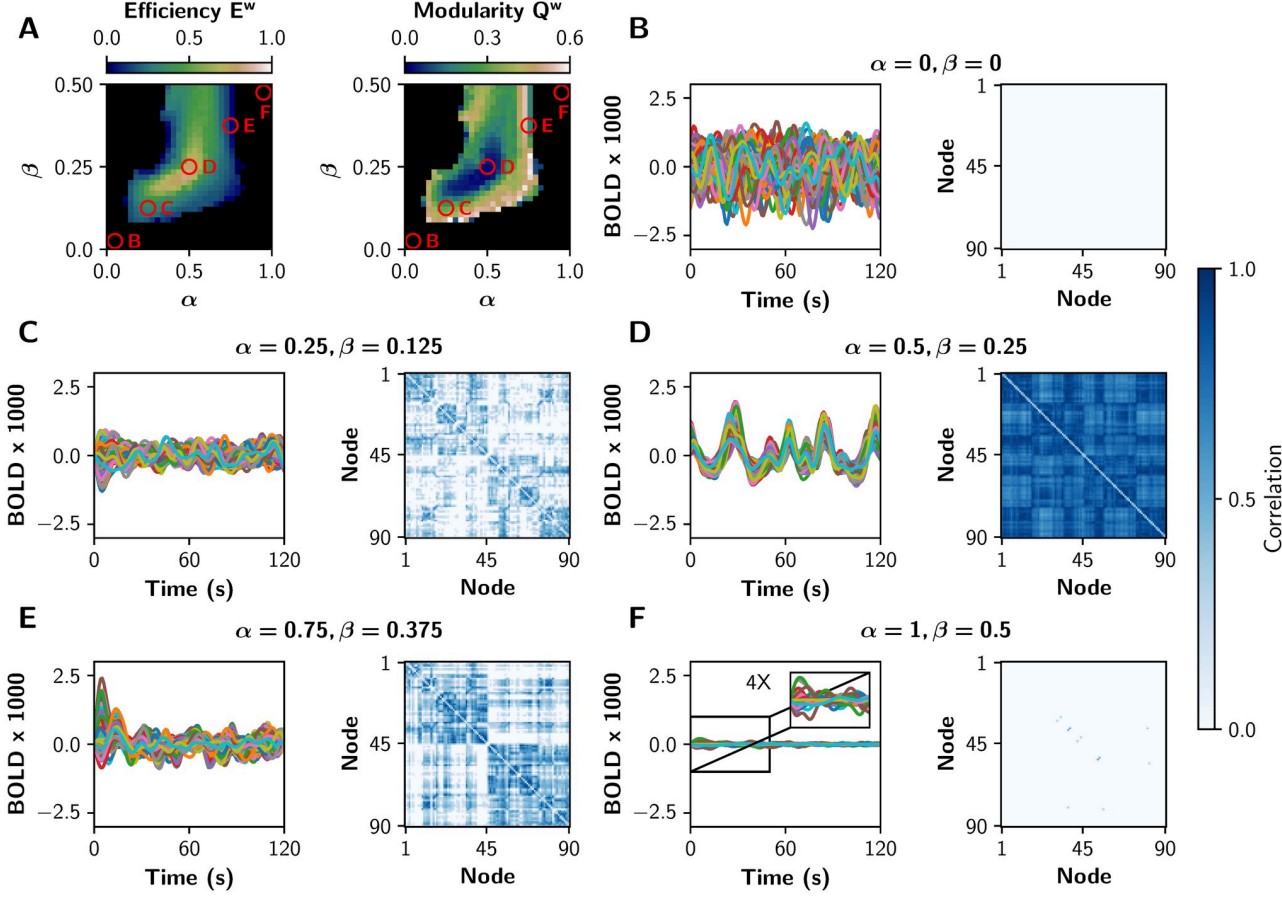

**Fig 3. fMRI-like sFCs at different values of $\alpha$ and $\beta$. A)** The red circles represent pairs of $(\alpha, \beta)$ values in which different integration/segregation profiles can be observed. **B-F)** BOLD-like signals, and their respective sFC matrices, for the $(\alpha, \beta)$ values shown in **A**. The sFC networks evolve from neither integration nor segregation (**B**, the nodes are disconnected), to a more integrated sFC (**C**). In **D** the integration is maximal, and a further increase of both parameters produces a more segregated sFC matrix (**E**). Finally, in **F** there is neither integration nor segregation (the pyramidal neurons are over-excited). We shown only 120 s of BOLD-like signals, while sFC matrices were built with the full-length time series (600 s).

the analysis previously performed in the $(\alpha, r_0)$ parameter space, but this time we changed $\beta$ alongside $\alpha$ following the relationship $\beta = 0.5\alpha$. The results are shown in S4 Fig. They are similar to those in Fig 4, but this time the relationship between $r_0$ and $E^w$ is no longer a sigmoid-like function, and instead it follows an inverted-U relationship.

## Changes in the EEG timescale match with the increase of integration

Previous experimental and theoretical works [13, 14, 28] suggest that neuromodulatory systems increase the signal-to-noise ratio, allowing neuronal populations to be sensitive to local or distant populations to a greater extent than noise. To test that, we measured the signal-to noise-ratio (SNR) using the power spectral density (PSD) function of each EEG-like signal (see Methods) and report the average value over all nodes. Additionally, we computed the average Kuramoto order parameter $\bar{R}$ [47], as a measure of global synchronization in the fast timescale of EEG. Values of $\bar{R}$ closer to 1 indicate a perfect in-phase synchronization, and values closer to 0 indicate complete asynchrony.

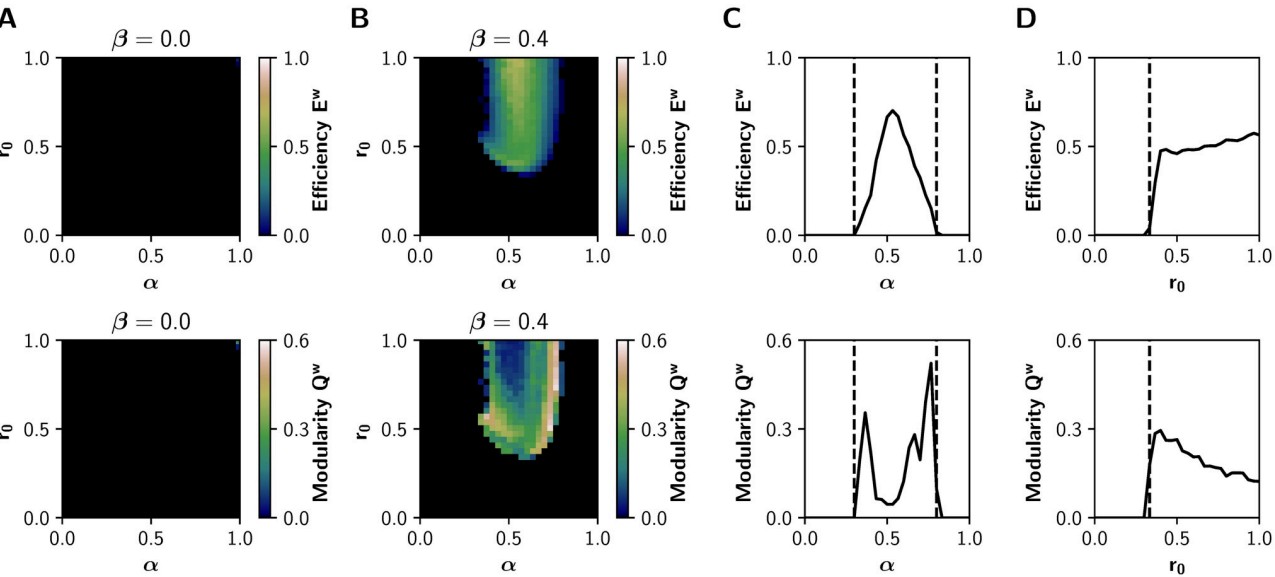

**Fig 4. Network features in the $(\alpha, r_0)$ parameter space. A-B)** Global efficiency $E^w$ (integration) and modularity $Q^w$ (segregation) of the graphs derived from the sFCs of the BOLD-like signals, for **A)** $\beta = 0$ (no action of the inhibitory gain) and **B)** $\beta = 0.4$. **C)** Transitions through the critical boundary in the $\alpha$ axis, with a fixed $r_0 = 1$ mV$^{-1}$ and $\beta = 0.4$. Critical transition points represented by black dashed lines at $\alpha = 0.3$ and $\alpha = 0.8$. **D)** Transitions in the $r_0$ axis, for a fixed $\alpha = 0.6$ and $\beta = 0.4$, with a critical transition at $r_0 = 0.33$ mV$^{-1}$.

Both SNR and $\bar{R}$ match the region of integration, measured as the global efficiency $E^w$ in the slowest BOLD timescale (Fig 5), supporting the idea that neuromodulatory systems promote integration by increasing SNR. In consequence, our results link in two different timescales the effect of neuromodulation in the coordination of brain activity. These results are not possible without the action of the inhibitory gain ($\beta = 0$, Fig 5A).

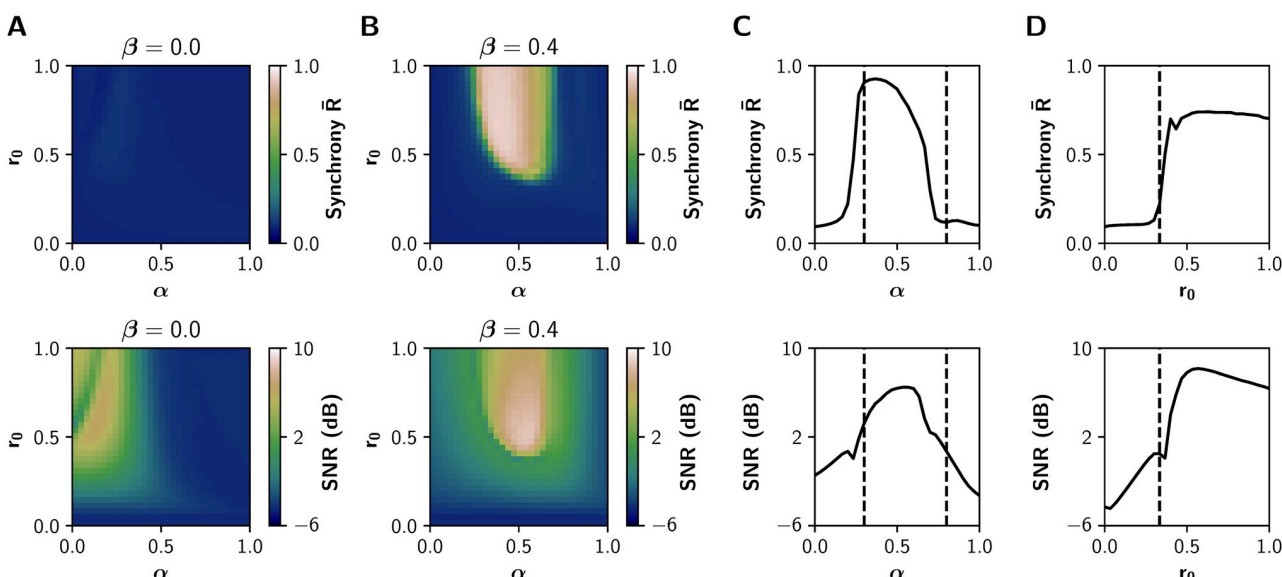

**Fig 5. EEG features in the $(\alpha, r_0)$ parameter space. A-B)** Average phase synchrony $\bar{R}$ and signal-to-noise ratio (SNR) measured from the EEG-like signals. **A)** No action of inhibitory gain ($\beta = 0$). **B)** The increase of $\bar{R}$ and the SNR matches with functional integration for $\beta = 0.4$. **C)** Transitions through the critical boundary in the $\alpha$ axis, with a fixed $r_0 = 1$ mV$^{-1}$. Critical transitions represented by black dashed lines at $\alpha = 0.3$ and $\alpha = 0.8$. **D)** Transitions in the $r_0$ axis, for a fixed $\alpha = 0.6$, with a critical transition at $r_0 = 0.33$ mV$^{-1}$.

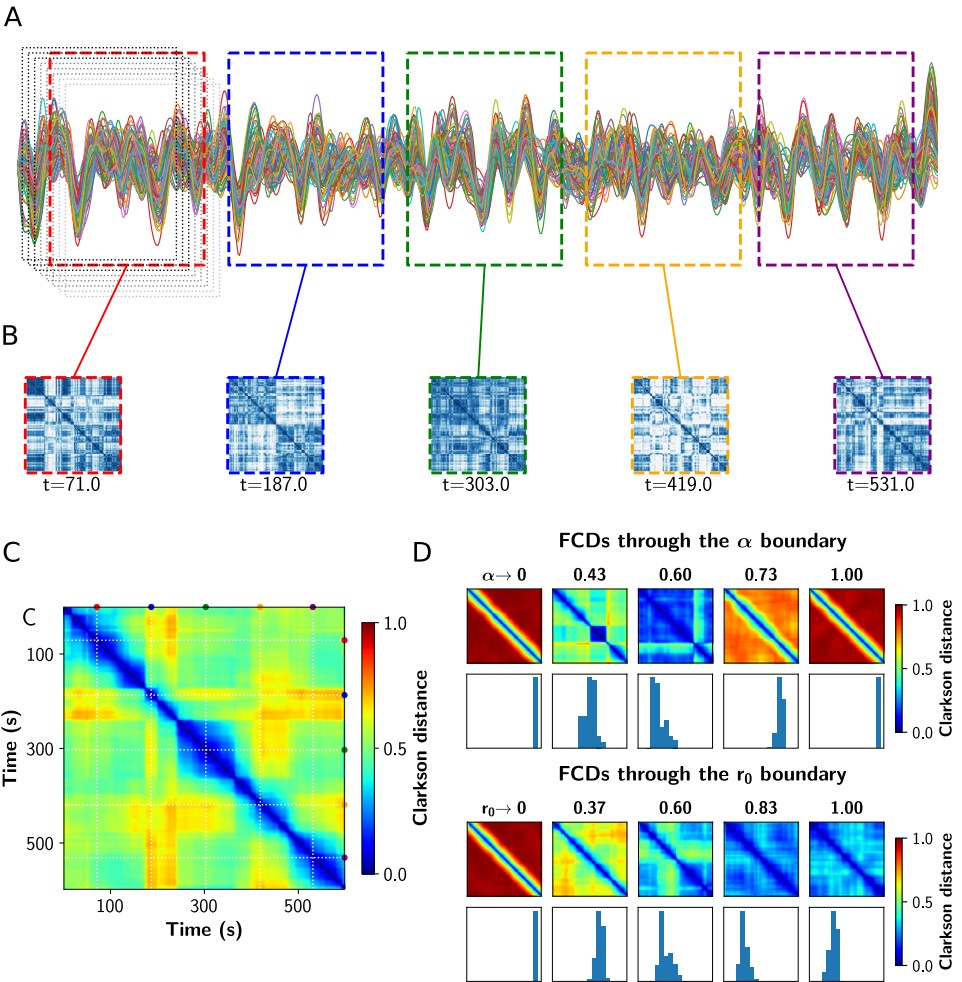

**Fig 6. Analysis of functional connectivity dynamics. A)** Sample fMRI BOLD time series showing the fixed length and overlapping time windows at the beggining. In color, the time windows corresponding to the FCs shown in B. **B)** FC matrices obtained in the colored time windows. **C)** Functional Connectivity Dynamics (FCD) matrix, where all the FCs obtained were vectorized and then compared against each other using a vector-based distance (Clarkson distance). **D)** FCD matrices through the critical boundary, in both $\alpha$ and $r_0$ direction. Below each FCD, a histogram of its upper triangular values is shown. The variance of these values constitutes a measure of multistability.

## Dynamical richness peaks near the critical boundary

As suggested by experimental [3] and computational studies [26], a shift to more segregated or integrated functional states decreases the topological variability of the network. Also, near the critical transitions for segregation to integration, network variability and communicability peak [26]. We tested this hypothesis performing a Functional Connectivity Dynamics (FCD) analysis [9, 10] on the BOLD-like signals, using the sliding windows approach depicted in Fig 6A–6C [48]. The resulting time vs time FCD matrix captures the concurrence of FC patterns, visualized as square blocks. We computed the variance of the FCD, var(FCD), as a multistability index [48], where values greater than 0 indicate the switching between different FC patterns. Additionally, we calculated the FCD speed $d_{typ}$ as described by Battaglia *et al.* [49], which captures how fast the FC patterns fluctuate over time. Values closer to 1 indicate a continuous change of diverse FC patterns, and closer to 0 the concurrence of stable and similar states over time.

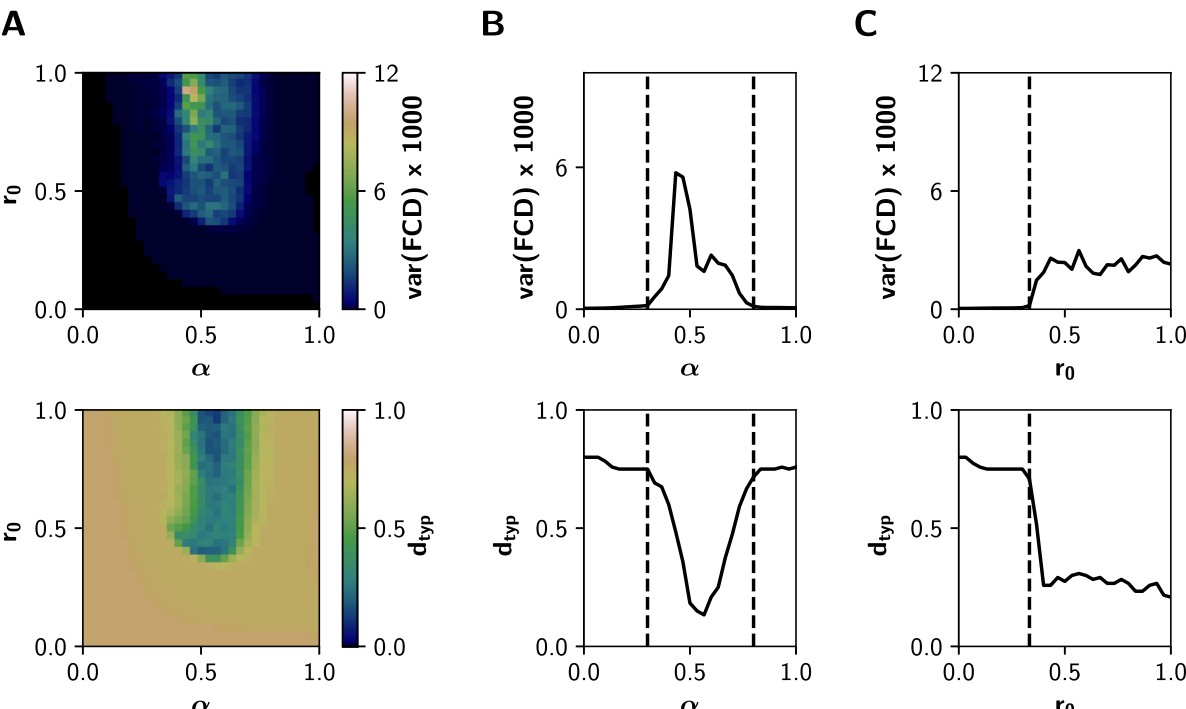

**Fig 7. Dynamical features of the system in the ($\alpha$, $r_0$) parameter space for $\beta$ = 0.4. A)** Multistability var(FCD) and typical FCD speed $d_{typ}$ measured from the Functional Connectivity Dynamics analysis (BOLD-like signals). **B)** Transitions through the critical boundary in the $\alpha$ axis, with a fixed $r_0$ = 1 mV$^{-1}$. Critical transitions represented by black dashed lines at $\alpha$ = 0.3 and $\alpha$ = 0.8. **C)** Transitions in the $r_0$ axis, for a fixed $\alpha$ = 0.6, with a critical transition at $r_0$ = 0.33 mV$^{-1}$.

In Fig 6D we show a set of FCD matrices obtained at different values of $\alpha$ and $r_0$, together with histograms of their off-diagonal values. Red FCD matrices (with high values) correspond to incoherent states, as the FC continuously evolves in time. On the other hand, a blue FCD matrix (with low values) indicates a fixed FC throughout the simulation. Multistability is higher for green/yellow patchy matrices, because this indicates FC patterns that change and also repeat over time. As can be inferred observing the FCD distributions, the *variance* of the values in the histograms –var(FCD)– can be used as a measure of multistability [48].

Fig 7 shows how multistability (var(FCD)) and FCD speed change in the whole ($\alpha$, $r_0$) space, for $\beta$ = 0.4. At low levels of both $\alpha$ and $r_0$, the neuronal activity is constituted mainly by noisy asynchronous signals, conditions associated to low (near 0) values of var(FCD), and with a high $d_{typ}$ (all FC patterns differ from each other, as expected for noise-driven signals) (Fig 7A). In the other extreme, for $r_0$ > 0.5 mV$^{-1}$ and $\alpha \in$ [0.5, 0.6], values that correspond to the integrated states, var(FCD) is also small and $d_{typ}$ falls close to 0. In consequence, integrated states are more stable and less susceptible to network reconfiguration over time. In contrast, var(FCD) peaks near the critical boundary, through the $\alpha$ and $r_0$ axes (Fig 7B and 7C). Moreover, crossing the boundaries is associated with a continuous decrease of $d_{typ}$: the emerging integration mediated by gain mechanisms is associated with more stable FC patterns over time.

## Discussion

In this work, we used a whole-brain neural mass model to investigate how local (meso-scale) neuromodulatory effects can impact the global functional network properties. Importantly, we studied the effect that the cholinergic system has in both, excitatory and inhibitory neurons,

along with the noradrenergic modulatory influence. Our model shows an increase in functional integration at intermediate values of the parameters that resemble the cholinergic and noradrenergic systems, following an inverted-U relation with the neuromodulation. In addition, the modulation of an intra-columnar inhibitory gain can promote functional integration and facilitates the effect of the other neuromodulatory systems. Finally, we show that our results hold for both the EEG and fMRI timescales, and that integration is accompanied by an increment in the signal-to-noise ratio, as well as a reduction of dynamical variability captured by the FCD analysis.

Our main motivation was to study the large-scale effects of the cholinergic neuromodulation of local inhibitory circuits. Although the cholinergic system increases the response of pyramidal neurons to external afferences through nicotinic and muscarinic receptors [34], the same system can promote intra-columnar inhibition, an effect mediated by nicotinic receptors expressed by inhibitory interneurons [28, 31, 50]. A possible consequence is the increase of the influence of external inputs, in comparison with the local intra-columnar inputs, shifting the flow of information from local to global processing. Based on these experimental findings [28, 31, 34, 50], we hypothesized that the cholinergic neuromodulation of the inhibitory interneurons facilitates functional integration. Our model shows that the action of the cholinergic system, on both the excitation of pyramidal neurons and the intra-columnar inhibitory feedback, can shift the system towards a functionally integrated regime. In this way, we propose a plausible biophysical mechanism of inhibitory-to-excitatory interneuron connection that facilitates functional integration of brain activity (see Fig 4).

The new intra-columnar connection that we introduced in the Jansen & Rit model –represented by the inhibitory gain $\beta$–, produces a higher dampening of the excitatory input to pyramidal neurons when their excitability is high. Conversely, when the pyramidal excitability decreases, the effect of the inhibitory loop between interneurons is low, and the excitatory loop can rise the excitability of pyramidal cells. In this way, the inhibitory gain provides a simple dynamical mechanism to homeostatically preserve the excitation/inhibition (E/I) balance at the node level. In contrast, a simple reduction of the excitatory feedback (e.g., decreasing $C_1$, S2 Fig), fails to compensate the E/I balance when pyramidal excitability is low, and thus has a limited ability to promote functional integration. This highlights the role of specific intra-columnar inhibitory feedback connections in shaping the network behavior, and justifies our modification of the model with a homeostatic mechanism. Similar types of inhibition-mediated control of the E/I balance have been implemented in a dynamic mean-field model [51] as well as in the Wilson-Cowan model [52]. Remarkably, the E/I balance has been considered a determinant element in the interplay between integration and segregation [53]

Our results also have an interpretation from the dynamical systems perspective. It has been proposed that, at rest, brain activity operates near a bifurcation point, where segregated (uncoordinated) and integrated (coordinated) regimes alternate in time. Then, a shift to more segregated or integrated states takes places with a change in behavioral context [5, 9, 10]. At the node level, the Jansen & Rit model has two Hopf supercritical bifurcations [54]. When $\alpha$ and $r_0$ are low, the node dynamics is defined by a stable focus (a fixed point with non-monotonic convergence), and thus pyramidal outputs consist of low amplitude noisy signals. Increasing both parameters causes the bifurcation into an unstable focus within a limit cycle, with high amplitude oscillations. Increasing $\alpha$ further produces a new bifurcation (into a stable focus) and the limit cycle disappears. The inhibitory gain $\beta$ constitutes a mechanism to keep the model working between the two bifurcations points, allowing the transitions between different functional states (more segregated or integrated). This again highlights the role of $\beta$ as an inhibitory control loop which preserves the E/I balance and sustains a richer brain dynamics.

Several clues suggest that the model we propose is in the right track. First, our model reproduces previous results of Shine *et al.*, also using a whole-brain model [26]: they reported and inverted-U relationship between cholinergic neuromodulation and integration, an increase in phase synchronization, and that integration is accompanied by a reduction in the time-resolved topological variability (captured, in our analysis, by the variance of the FCD). Second, functional integration matches with an increase of SNR, a common effect attributed to neuro-modulatory systems [13, 14, 28, 55]. Third, integration is accompanied in our model by a reduction in oscillatory frequency –which falls within the Theta range of EEG spectrum– (S8 Fig); an effect that is also perceived in several cognitive tasks [56, 57], and reproduced by other computational studies [58, 59].

From an experimental point of view, an increase in the local and global efficiency (integration) in fMRI has been reported after the administration of nicotine both in resting state conditions [37] and during an attentional task [38]. Interestingly, the performance was positively correlated with the global efficiency, and negatively correlated with the average clustering coefficient. Some nicotinic agonists have pro-cognitive effects as well, in health and disease [60]. Considering the relationship between functional integration and cognition [2, 3, 13], our model suggests that the possible pro-cognitive effects associated with the cholinergic system are due to a selective increase in the excitability of excitatory and inhibitory neural populations within brain areas. Thus, our computational approach –in the same spirit as Wylie *et al.* [37] and Gießing *et al.* [38]– links the meso-scale consequences of inhibitory interneurons neuro-modulation with the functional network topology features at the whole-brain level.

On the other hand, the inverted-U relationship between neuromodulation and integration that we are reporting in our whole-brain model, has not been experimentally observed. However, an inverted-U function between cholinergic and noradrenergic neuromodulation and in-task performance [15, 27]; as well as between in-task performance and functional integration [12] has been reported. Taking these together we propose, in accordance to Shine *et al.*. [13, 26], that neuromodulation improves cognitive performance by boosting integration. The results from computational modeling should nevertheless be verified by measuring both integration and cognitive performance as functions of neuromodulatory activity, e.g., as functions of the dose of a cholinergic/noradrenergic drug.

There is a lot of room for further progress starting from this work. Future research may consider the addition of neuromodulatory maps [41, 61] in order to take into account the heterogeneous expression of the receptors, or explore models that can reproduce the effect of other neuromodulatory systems [17]. Furthermore, it is known that cholinergic and noradrenergic projections have a specific spatial organization [62, 63]. Our model considers neuromodulation to be static, that is, the parameters $\alpha$, $\beta$ and $r_0$ do not change over time, as in tonic neuromodulation. An improvement to our model may be the addition of the release and reuptake dynamics of neuromodulators, as in Kringelbach *et al.* [64] or the characterization of the dynamics under acute neuromodulatory 'pulses'. Other interneurons subtypes and their modulation could be included –such as fast-spiking inhibitory interneurons– to account for the faster EEG features of brain activity [65]. Finally, the graph theoretical analysis used here only considers pairwise interactions, neglecting high-order effects that may contain important information about high dimensional functional brain interactions. Information-theoretical [66, 67] and algebraic topological approaches [68–70] may provide complementary insights of high-order interdependencies in the brain.

Our findings shed light on a better understanding of neurophysiological mechanisms involved in the functional integration and segregation of the human brain activity and constitutes a step forward from the neuromodulatory framework proposed by Shine [13], including the role of a second cholinergic target and also highlighting the role of a homeostatic inhibitory

feedback. This line of research may have plentiful of scientific and clinical implications, as a vast body of evidence suggest that functional integration and segregation may be altered in neuropsychiatric disorders [53, 71, 72], e.g, in Alzheimer disease and Attention-Deficit/Hyper-activity Disorder (ADHD) [73, 74]. These results point out the usefulness of graph theoretical analysis to exhibit functional markers for characterizing and understanding neuropsychiatric disorders. Understanding the neuromodulatory mechanisms that underlie the imbalances of integration and segregation will lead to a more profound understanding of how the brain works in health and disease and to future progress in pharmacological treatments.

## Methods

### Whole-brain neural mass model

To simulate neuronal activity we used a modified version of the Jansen & Rit neural mass model [35, 36]. In this model, a cortical column consists of a population of pyramidal neurons (that reside in cortical column layer V) with projections to other two populations: excitatory interneurons (nearby pyramidal cells which reside in the same layer than the principal pyramidal population) and inhibitory interneurons; both project back to the pyramidal population. The dynamical evolution of the three populations within the cortical column is modeled by two blocks each. The first transforms the average pulse density in average post-synaptic membrane potential (which can be either excitatory or inhibitory) (Fig 1A). This block, denominated post synaptic potential (PSP) block, is represented by an impulse response function

$$h_E(t) = \begin{cases} Aate^{-at}, & t \geq 0 \\ 0, & t < 0 \end{cases} \tag{1}$$

for the excitatory outputs, and

$$h_I(t) = \begin{cases} Bbte^{-bt}, & t \geq 0 \\ 0, & t < 0, \end{cases} \tag{2}$$

for the inhibitory ones. The constants $A$ and $B$ define the maximum amplitude of the PSPs for the excitatory (EPSPs) and inhibitory (IPSPs) cases respectively, while $a$ and $b$ represent the inverse time constants for the excitatory and inhibitory postsynaptic action potentials, respectively. The second block transforms the postsynaptic membrane potential in average pulse density, and is given by a sigmoid function of the form

$$S(v, r) = \frac{\zeta_{max}}{1 + e^{r(\theta - v)}}, \tag{3}$$

with $\zeta_{max}$ as the maximum firing rate of the neuronal population, $r$ the slope of the sigmoid function, and $\theta$ the half maximal response of the population.

Additionally, the pyramidal neurons receive an external stimulus $p(t)$, whose values were taken from a Gaussian distribution with mean $\mu = 2$ impulses/s and standard deviation $\sigma = 2$. Different values of $\sigma$ were explored; qualitatively the results are similar for different $\sigma$ values, but the magnitude of integration decreases with $\sigma$. This exploration is shown in the S5 Fig. In the same manner, the mean of the Gaussian distribution $\mu$ has an effect in decreasing the synchronization and integration, as shown in the S5 Fig.

To study the effect of the neuromodulatory systems at the macro-scale level, we included long-range pyramidal-to-pyramidal neurons and short-range inhibitory-to-excitatory

interneurons couplings, to mimic the effects of neuromodulation through the excitatory and inhibitory gain parameters, respectively. This short-range coupling between interneurons, well described at the meso-scale level [39, 40], constitutes a modification of the original equations. A bifurcation analysis of the model at different values of $\beta$ reveals that this parameter shifts the oscillatory regime of the system towards larger values of the external input $p(t)$ (S7 Fig). Nevertheless, the main oscillatory rhythm of the model is well maintained within the $\alpha$ band frequency (around 10 Hz).

In the model presented in the Fig 1A, each node $i \in [1 . . . N]$ represents a single brain area. The nodes are connected by a normalized structural connectivity matrix $\widetilde{M}$ (Fig 1B). This matrix is derived from a human connectome [41] parcellated in $n = 90$ cortical and subcortical regions with the automated anatomical labelling (AAL) atlas [42]; the matrix is undirected and takes values between 0 and 1. Because long-range connections are mainly excitatory [43, 44], only links between the pyramidal neurons of a node $i$ with pyramidal neurons of a node $j$ are considered. We applied a local normalization procedure to the structural connectivity matrix $M$. The normalization consisted in dividing all the columns belonging to a node $i$ by the in-strength of the node. The entries of the resulting normalized matrix $\widetilde{M}$ are defined as

$$\widetilde{M}_{ij} = \frac{M_{ij}}{\sum_{j=1, j\neq i}^{n} M_{ij}} \qquad (4)$$

The local normalization procedure constitutes a form of homeostatic plasticity, which equalizes the excitatory inputs that the nodes receive, while preserving the structural topology. It has been reported that this mechanism improves the fit of a whole-brain mesoscopic model to empirical fMRI data, and leads to a better estimation of the functional connectivity [75].

The overall set of equations, for a node $i$, includes the within and between nodes activity

$$
\begin{aligned}
\dot{x}_{0,i}(t) =\ & y_{0,i}(t) \\
\dot{y}_{0,i}(t) =\ & Aa\left[S(C_2 x_{1,i}(t) - C_4 x_{2,i}(t) + C\alpha z_i(t), r_0)\right] - 2ay_{0,i}(t) - a^2 x_{0,i}(t) \\
\dot{x}_{1,i}(t) =\ & y_{1,i}(t) \\
\dot{y}_{1,i}(t) =\ & Aa\left[p(t) + S(C_1 x_{0,i}(t) - C\beta x_{2,i}(t), r_1)\right] - 2ay_{1,i}(t) - a^2 x_{1,i}(t) \\
\dot{x}_{2,i}(t) =\ & y_{2,i}(t) \\
\dot{y}_{2,i}(t) =\ & Bb\left[S(C_3 x_{0,i}(t), r_2)\right] - 2by_{2,i}(t) - b^2 x_{2,i}(t) \\
\dot{x}_{3,i}(t) =\ & y_{3,i}(t) \\
\dot{y}_{3,i}(t) =\ & A\bar{a}\left[S(C_2 x_{1,i}(t) - C_4 x_{2,i}(t) + C\alpha z_i(t), r_0)\right] - 2\bar{a}y_{3,i}(t) - \bar{a}_i^2 x_{3,i}(t)
\end{aligned}
\qquad (5)
$$

where $x_0$, $x_1$, $x_2$ correspond to the outputs of the PSP blocks of the pyramidal neurons, and excitatory and inhibitory interneurons, respectively, and $x_3$ the long-range outputs of pyramidal neurons. The constants $C_1$, $C_2$, $C_3$ and $C_4$ scale the connectivity between the neural populations (see Fig 1A). The first pair of equations, $x_0$ and $y_0$, are related to the outputs of pyramidal cells to both interneurons; the second pair, $x_1$ and $y_1$, represent all the local excitatory inputs that the pyramidal neurons receive; $x_2$ and $y_2$ constitute the inhibitory contribution to pyramidal cells; and finally, $x_3$ and $y_3$ correspond to the long-range excitatory outputs of pyramidal neurons. We used the original parameter values of Jansen & Rit [35, 36]: $\zeta_{max} = 5$ s$^{-1}$, $\theta = 6$

mV, $r_0 = r_1 = r_2 = 0.56$ mV$^{-1}$, $a = 100$ s$^{-1}$, $b = 50$ s$^{-1}$, $A = 3.25$ mV, $B = 22$ mV, $C_1 = C$, $C_2 = 0.8C$, $C_3 = 0.25C$, $C_4 = 0.25C$, and $C = 135$. The parameters $A$, $B$, $a$ and $b$ were selected as in the original Jansen & Rit model [35, 36] to produce IPSPs longer in amplitude and latency in comparison with the EPSPs. The inverse of the characteristic time constant for the long-range EPSPs was defined as $\bar{a} = 0.5a$. This choice was based on the fact that long-range excitatory inputs of pyramidal neurons target their apical dendrites, and consequently this decreases the time course of the EPSPs at the soma due to dendritic nonlinearities and a gradient of input impedances [76].

The overall input from other cortical columns $j \neq i$ to the column $i$ is given by

$$z_i(t) = \sum_{j=1, j \neq i}^{n} \widetilde{M}_{ij} x_{3,j}(t) \tag{6}$$

The average PSP of pyramidal neurons in column $i$ characterizes the EEG-like signal in the source space; it is computed as [35, 36]

$$v_i(t) = C_2 x_{1,i}(t) - C_4 x_{2,i}(t) + C\alpha z_i(t) \tag{7}$$

The firing rates of pyramidal neurons $\zeta_i(t) = S(v_i(t), r_0)$ were used to simulate the fMRI BOLD recordings. The parameters $\alpha$, $\beta$ and $r_0$ account for the influence of the neuromodulatory systems (Fig 1C), as described in next subsection.

## Neuromodulation

The effects of the cholinergic system were modeled by the parameters $\alpha$ and $\beta$. The parameter $\alpha$ increases the long-rage pyramidal to pyramidal neuron coupling through the $\widetilde{M}$ matrix. Although $\alpha$ does not control directly the excitability, increasing $\alpha$ amplifies the input to pyramidal neurons [13, 14]. The parameter $\beta$ scales the short-range inhibitory-to-excitatory interneurons coupling, decreasing the recurrent excitation to pyramidal neurons [28]. We refer to $\alpha$ as the excitatory gain, and $\beta$ as the inhibitory gain. In comparison with the current framework proposed by Shine [13], the novelty of our neuromodulatory approach is the inclusion of the inhibitory gain to the model. The effect of the noradrenergic system, designated as filter gain, was simulated controlling the parameter $r_0$, which represents the sigmoid function slope of the pyramidal population, and increases the signal-to-noise ratio of pyramidal cells [14, 25].

## Simulation

Following Birn *et al.* [77], we ran simulations to generate the equivalent of 11 min real-time recordings, discarding the first 60 s. The system of differential equations (Eq (5)) was solved with the Euler–Maruyama method, using an integration step of 1 ms. We used six random seeds which controlled the initial conditions and the stochasticity of the simulations. We simulated neuronal activity sweeping the parameters $\alpha \in [0, 1]$, $\beta \in [0, 0.5]$ and $r_0 \in [0, 1]$. All the simulations were implemented in Python and the codes are freely available at https://github.com/vandal-uv/Neuromod2020.

## Simulated fMRI BOLD signals

We used the firing rates $\zeta_i(t)$ to simulate BOLD-like signals from a generalized hemodynamic model [45]. An increment in the firing rate $\zeta_i(t)$ triggers a vasodilatory response $s_i$, producing blood inflow $f_i$, changes in the blood volume $v_i$ and deoxyhemoglobin content $q_i$. The

corresponding system of differential equations is

$$\dot{s}_i(t) = \zeta_i(t) - \frac{s_i(t)}{\tau_s} - \frac{f_i(t) - 1}{\tau_f}$$

$$\dot{f}_i(t) = s_i(t)$$

$$\dot{v}_i(t) = \frac{f_i(t) - v_i(t)^{1/\kappa}}{\tau_v} \tag{8}$$

$$\dot{q}_i(t) = \frac{\dfrac{f_i(t)(1 - (1 - E_0)^{1/f_i(t)})}{E_0} - \dfrac{q_i(t)v_i(t)^{1/\kappa}}{v_i(t)}}{\tau_q},$$

where $\tau_s$, $\tau_f$, $\tau_v$ and $\tau_q$ represent the time constants for the signal decay, blood inflow, blood volume and deoxyhemoglobin content, respectively. The stiffness constant (resistance of the veins to blood flow) is given by $\kappa$, and the resting-state oxygen extraction rate by $E_0$. Finally, the BOLD-like signal of node $i$, denoted $B_i(t)$, is a non-linear function of $q_i(t)$ and $v_i(t)$

$$B_i(t) = V_0\left[k_1(1 - q_i(t)) + k_2\left(1 - \frac{q_i(t)}{v_i(t)}\right) + k_3(1 - v_i(t))\right] \tag{9}$$

where $V_0$ represent the fraction of venous blood (deoxygenated) in resting-state, and $k_1$, $k_2$, $k_3$ are kinetic constants. We used the same parameters as in Stephan *et al.* [45]: $\tau_s = 0.65$, $\tau_f = 0.41$, $\tau_v = 0.98$, $\tau_q = 0.98$, $\kappa = 0.32$, $E_0 = 0.4$, $k_1 = 2.77$, $k_2 = 0.2$, $k_3 = 0.5$.

The system of differential equations (Eq (8)) was solved with the Euler method, using an integration step of 1 ms. The signals were band-pass filtered between 0.01 and 0.1 Hz with a 3rd order Bessel filter. These BOLD-like signals were used to build the functional connectivity (FC) matrices from which the subsequent analysis of functional network properties is performed using tools from graph theory.

Although the nodes consist of three neural masses, there is some evidence that the hemodynamic response is related mainly to excitatory activity [78]. In fact, some reports suggest that inhibitory activity does not trigger a measurable BOLD response, because the inhibitory connections are relatively few and their energy expenditure is lower [79]. Nevertheless, we reproduced the Fig 2 using a combined BOLD response, and we found no noticeable differences (see S6 Fig).

## Global phase synchronization

As a measure of global synchronization in the EEG timescale, we calculated the Kuramoto order parameter $R(t)$ [47] of the EEG-like signals $v(t)$ derived from the Jansen & Rit model. First, the raw signals were filtered with a 3rd order Bessel band-pass filter using their frequency of maximum power (usually between 4 and 10 Hz) ±3 Hz. Then, the instantaneous phase $\phi(t)$ was obtained with the Hilbert transform.

The global phase synchrony is computed as:

$$\bar{R} = \langle |\langle e^{j\phi_i(t)}\rangle_N|\rangle_t \tag{10}$$

where $\phi_i(t)$ is the phase of the oscillator $i$ over time, $j = \sqrt{-1}$ the imaginary unit, $|\bullet|$ denotes the module, $\langle\rangle_N$ denotes the average over all nodes, and $\langle\rangle_t$ the average over time. A value of $\bar{R}$ equal to 1 indicates perfect in-phase synchronization of all the set $N$ of oscillators, while a value equal to 0 indicates total asynchrony.

## Signal-to-noise ratio

We measured the average signal-to-noise ratio (SNR) over all raw signals and nodes, using the power spectral density function denoted $PSD(\omega)$. This function was calculated using the Welch's method [80], with 20 s time windows overlapped by 50%. We excluded the 2nd to 5th harmonics [81]. For a node $i$, the signal power, $P_{signal}$, was measured as the area under the curve of $PSD(\omega)$ within $\omega_i \pm 1Hz$. Noise power, $P_{noise}$, corresponds to the area under the curve of $PSD(\omega)$ outside the $\pm 1Hz$ window. Then, the SNR was calculated as

$$SNR = 10 \log_{10} \frac{P_{signal}}{P_{noise}}, \tag{11}$$

The SNR was computed for each node $i$ and we reported the average over all nodes.

## Functional connectivity and graph thresholding

The static Functional Connectivity (sFC) matrices were built from pairwise Pearson's correlations of the entire BOLD-like time series. Instead of employing an absolute or proportional thresholding, we thresholded the sFC matrices using Fourier transform (FT) surrogate data [82] to avoid the problem of introducing spurious correlations [83]. The FT algorithm uses a phase randomization process to destroy pairwise correlations, preserving the spectral properties of the signals (the surrogates have the same power spectrum as the original data). We generated 500 surrogates time series of the original set of BOLD-like signals, and then built the surrogates sFC matrices. For each one of the $(n^2 - n)/2$ possible connectivity pairs (with $n = 90$) we fitted a normal distribution of the surrogate values. Using these distributions we tested the hypothesis that a pairwise correlation is higher than chance (that is, the value is at the right of the surrogate distribution). To reject the null hypothesis, we selected a $p$-value equal to 0.05, and corrected for multiple comparisons with the FDR Benjamini-Hochberg procedure [84] to decrease the probability of make type I errors (false positives). The entries of the sFC matrix associated with a $p$-value greater than 0.05 were set to 0. The result is a thresholded, undirected, and weighted (with only positive values) sFC matrix.

## Integration and segregation

Integration and segregation were evaluated over the thresholded sFC matrices. We employed the weighted versions of transitivity [85] and global efficiency [86] to measure segregation and integration, respectively. A detailed description of the metrics used can be found in Rubinov & Sporns [46]. The transitivity (similar to the average clustering coefficient) counts the fraction of triangular motifs surrounding the nodes (the equivalent of counting how many neighbors are also neighbors of each other), with the difference that it is normalized collectively. It is defined as

$$T^w = \frac{\sum_{i \in N} 2t_i^w}{\sum_{i \in N} k_i^w (k_i^w - 1)}, \tag{12}$$

being $N$ the set of all nodes of the network with $n$ number of nodes, $t_i^w$ the geometric average of the triangles around the node $i$, and $k_i^w$ the node weighted degree. The supra-index $w$ is used to refer to the weighted versions of the topological network measures. On the other hand, the global efficiency is a measure of integration based on paths over the graph: it is defined as the

inverse of the average shortest path length. This metric is computed as

$$E^w = \frac{1}{n} \sum_{i \in N} \frac{\sum_{j \in N, j \neq i} (d_{ij}^w)^{-1}}{n - 1},$$  (13)

where $d_{ij}^w$ is the shortest path between the nodes $i$ and $j$.

We also calculated other two measures of integration and segregation: the participation coefficient $PC^w$ and modularity $Q^w$, respectively, both based on the detection of the network's communities [46]. The detection of so-called communities or network modules in the thresholded sFC matrix, was based on the Louvain's algorithm [87, 88]. The algorithm assigns a module to each node in a way that maximizes the modularity (14). We used the weighted version of the modularity [89] defined as

$$Q^w = \frac{1}{l^w} \sum_{i,j \in N} \left[ w_{ij} - \frac{k_i^w k_j^w}{l^w} \right] \delta_{m_i, m_j}$$  (14)

where $w_{ij}$ is the weight of the link between $i$ and $j$, $l^w$ is the total number of weighted links of the network, $m_i$ ($m_j$) the module of the node $i$ ($j$). The Kronecker delta $\delta_{m_i, m_j}$ is equal to 1 when $m_i = m_j$ (that is, when two nodes belongs to the same module), and 0 otherwise. Because the Louvain's algorithm is stochastic, we employed the consensus clustering algorithm [90]. We ran the Louvain's algorithm 200 times with the resolution parameter set to 1.0 (this parameter controls the size of the detected modules; larger values of this parameter allows the detection of smaller modules). Then, we built an agreement matrix $G$, in which an entry $G_{ij}$ indicates the proportion of partitions in which the pairs of nodes $(i, j)$ share the same module (so, the entries of $G$ are bounded between 0 and 1). Then, we applied an absolute threshold of 0.5 to the matrix $G$, and ran again the Louvain's algorithm 200 times using $G$ as input, producing a new consensus matrix $G'$. This last step was repeated until convergence to an unique partition.

Finally, we computed the weighted version of the participation coefficient [91]. This metric quantifies, for each individual node, the strength of between-module connections respect to the within-module connections, and is defined as

$$\langle PC^w \rangle_N = \frac{1}{n} \sum_{i \in N} PC_i^w = \frac{1}{n} \sum_{i \in N} \left( 1 - \sum_{m \in \mathcal{M}} \left( \frac{k_i^w(m)}{k_i^w} \right)^2 \right)$$  (15)

where $PC_i^w$ is the weighted participation coefficient for the node $i$, and $\langle PC^w \rangle_N$ is the average overall nodes. The functional network analysis was done in Python using the Brain Connectivity Toolbox [46].

## Functional connectivity dynamics

The FCD matrix captures the evolution of FC patterns and, consequently, the dynamical richness of the network [9, 10]. We used the sliding window approach [9, 48] depicted in Fig 6. Window length was set to 100 s with a displacement of 2 s between consecutive windows (Fig 6A). The length was chosen on the basis of the lower limit of the band-pass filter (0.01 Hz), in order to minimize spurious correlations [92]. For each window, an FC matrix was calculated from the pairwise Pearson's correlations of BOLD-like signals (neglecting negative values), thus we obtained 251 weighted and undirected FC matrices from the 600 s simulated BOLD-like signals (Fig 6B).

The upper triangle of each FC matrix is unfolded to make a vector, and the FCD is built by calculating the Clarkson angular distance $\lambda(x, y) = \frac{1}{\sqrt{2}} \left\| \frac{x}{\|x\|} - \frac{y}{\|y\|} \right\|$ [93] between each pair of

FCs (Fig 6C)

$$FCD_{ij} = \lambda(FC(t_i), FC(t_j)) \tag{16}$$

The variance of the values in the upper triangle of the FCD, with an offset of $\tau = 100$ s from the diagonal (e.g., the variance of the histograms of Fig 6D), is taken as a measure of dynamical richness [48].

The speed of the FCD was measured as described by Battaglia *et al.* [49]. We computed the histogram of FCD values through a straight line from FCD$(\tau, 0)$ to FCD$(t_{max}, t_{max} - \tau)$, with $t_{max} = 600$ s as the total time-length of the signals and $\tau = 100$ s. The median of the histogram distribution corresponded to the typical FCD speed $d_{typ}$. Values closer to 1 indicate a constant switching of states, and values closer to 0 correspond to stable FC patterns.

## Supporting information

**S1 Fig. Alternative measures of network segregation and integration in the $(\alpha, \beta)$ parameter space. A)** Mean participation coefficient $PC^w$ (integration) and transitivity $T^w$ (segregation). **B-C)** Transitions in the $\alpha$ and $\beta$ axes. Dashed lines represent critical transitions.
(PDF)

**S2 Fig. Network features in the $(\alpha, C_1)$ parameter space. A)** Global efficiency $E^w$ (integration) and modularity $Q^w$ (segregation) of the graphs derived from the sFCs of the BOLD-like signals. **B)** Transitions in the $\alpha$ axis, for a fixed $C_1 = 0$. **C)** Transitions in the $C_1$ axis, for a fixed $\alpha = 0.5$. Dashed lines represent critical transitions.
(PDF)

**S3 Fig. Alternative measures of network segregation and integration in the $(\alpha, r_0)$ parameter space for $\beta = 0.4$. A-B)** Mean participation coefficient $PC^w$ (integration) and transitivity $T^w$ (segregation) with **A)** $\beta = 0$ and **B)** $\beta = 0.4$. **C-D)** Transitions in the $\alpha$ and $r_0$ axes. Dashed lines represent critical transitions.
(PDF)

**S4 Fig. Simultaneous effect of $\alpha$, $\beta$ and $r_0$ in network features. A)** Global efficiency $E^w$ (integration) and modularity $Q^w$ (segregation) of the graphs derived from the sFCs of the BOLD-like signals. **B)** Transitions in the $\alpha$ axis, for a fixed $r_0 = 0.5$ mV$^{-1}$. **C)** Transitions in the $r_0$ axis, for a fixed $\alpha = 0.5$. Dashed lines represent critical transitions. Both coupling parameters change in parallel following the relationship $\beta = 0.5\alpha$.
(PDF)

**S5 Fig. Effect of the input standard deviation, $\sigma$, and mean, $\mu$, in synchronization and integration. A-B)** Average phase synchrony $\bar{R}$ measured over the EEG-like signals, and global efficiency $E^w$ computed over the sFC matrices build using the BOLD-like signals. $\sigma$ and $\mu$ reduces the increment of $\bar{R}$ and $E^w$ mediated by $\alpha$ and $\beta$. Both coupling parameters change in parallel following the relationship $\beta = 0.5\alpha$. **C-D)** Phase synchrony $\bar{R}$ and global efficiency $E^w$ as a function of $\alpha$ and $\beta$, for different $\sigma$ (left hand) and $\mu$ (right hand) values.
(PDF)

**S6 Fig. Network integration computed from mixed BOLD-like signals. A)** Global efficiency $E^w$ computed in the entire $(\alpha, \beta)$ parameter space. BOLD-like signals were computed using only the firing rates of pyramidal neurons. **B)** $E^w$ calculated starting from a summation of the BOLD-like signals simulated using the firing rates of the three neural masses: pyramidal neurons, excitatory and inhibitory interneurons. **C)** Difference in the global efficiency $\Delta E^w$ between the two matrices in the $(\alpha, \beta)$ parameter space. Green values correspond to

near-zero difference between the matrices. There is not a noticeable difference between them.
(PDF)

**S7 Fig. Bifurcation analysis of the modified Jansen & Rit model and comparison to the original model.** Bifurcations of a single-node model with respect to the mean external input parameter $p$, at three values of $\beta$. Note that $\beta = 0$ corresponds to the original Jansen & Rit model. The value depicted in the $y$-axis is the variable $x_0$. Red and black lines denote stable and unstable fixed points, respectively. Green solid lines and blue dashed lines represent stable and unstable periodic attractors, respectively (denoting the maximum and minimum values of the oscillation). At the right, sample time courses (3 seconds) of the EEG-like signal ($C_2 x_1(t) - C_4 x_2(t)$) at three values of $p$, and their corresponding power spectra below.
(PDF)

**S8 Fig. Effect of neuromodulation on the mean oscillatory frequency $\omega$.** The frequency falls in the Theta range (4-8 Hz) of the EEG spectrum in both the **A** ($\alpha, \beta$) and **B** ($\alpha, r_0$) parameter spaces. In the first case we fixed $r_0 = 0.56$ mV$^{-1}$, and in the second one $\beta = 0.4$. The frequency drop off matches with phase synchronization (see Fig 5).
(PDF)

## Acknowledgments

We thank to Gustavo Deco who kindly provided the anatomical connectivity matrix used in the model. We also want to thank to Chiayu Chiu and Andrés Chávez for their feedback and suggestions about the manuscript.

## Author Contributions

**Conceptualization:** Carlos Coronel-Oliveros, Rodrigo Cofré, Patricio Orio.

**Data curation:** Carlos Coronel-Oliveros.

**Formal analysis:** Carlos Coronel-Oliveros, Patricio Orio.

**Funding acquisition:** Carlos Coronel-Oliveros, Rodrigo Cofré, Patricio Orio.

**Investigation:** Carlos Coronel-Oliveros.

**Methodology:** Carlos Coronel-Oliveros, Rodrigo Cofré, Patricio Orio.

**Project administration:** Patricio Orio.

**Resources:** Patricio Orio.

**Supervision:** Rodrigo Cofré, Patricio Orio.

**Validation:** Patricio Orio.

**Visualization:** Carlos Coronel-Oliveros.

**Writing – original draft:** Carlos Coronel-Oliveros, Rodrigo Cofré, Patricio Orio.

**Writing – review & editing:** Carlos Coronel-Oliveros, Rodrigo Cofré, Patricio Orio.

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
