## [Decision Letter · Decision Letter 0]

28 Oct 2020

Dear Mr. Orio,

Thank you very much for submitting your manuscript "Cholinergic neuromodulation of inhibitory interneurons facilitates functional integration in whole-brain models" for consideration at PLOS Computational Biology.

As with all papers reviewed by the journal, your manuscript was reviewed by members of the editorial board and by several independent reviewers. In light of the reviews (below this email), we would like to invite the resubmission of a significantly-revised version that takes into account the reviewers' comments.

The major points to address range from the situation in the state of the art, to the nature and clarity of the proposed approach, to the application and implications, so we suggest extra care in organically addressing all of them.

We cannot make any decision about publication until we have seen the revised manuscript and your response to the reviewers' comments. Your revised manuscript is also likely to be sent to reviewers for further evaluation.

Sincerely,

Daniele Marinazzo

Deputy Editor

PLOS Computational Biology

Daniele Marinazzo

Deputy Editor

PLOS Computational Biology

Reviewer's Responses to Questions

**Comments to the Authors:**

Reviewer #1: Thank you for inviting me to review this manuscript by Coronel-Oliveros and colleagues, in which the authors adapt a Jansen-Rit neuronal model to replicate and extend previous work relating the ascending neuromodulatory arousal system to network-level topological characteristics and temporal signatures of neural activity.

• Can the authors please discuss what the ‘excitatory interneuron’ population represents in the Jansen-Rit model. The term ‘interneuron’ is typically used to describe locally-projecting GABAergic neurons (doi.org/10.1146/annurev-neuro-070918-050421). Although some of these neurons, by virtue of their inhibitory projections to other GABAergic neurons, are thought to disinhibit excitatory pyramidal neurons (e.g., doi.org/10.1038/s41593-019-0508-y), it is hard to know whether this is the population the authors are referring to. Alternatively, the excitatory interneuron population could reflect local (i.e., within column) recurrent activity, however this would imply a different set of properties (e.g., time-scales) and responsiveness (or not) to different classes of neuromodulatory neurotransmitters (see next point).

• Along these lines, I worry that the link between the cholinergic system and the inhibitory connection introduced in their model may not be as specific as the authors hope. Importantly, this depends on precisely how the authors (and others) conceptualize the excitatory interneuron population. If they are conceptualizing EI as VIP+ interneurons, then these cells have demonstrated responsivity to serotonin (doi.org/10.1523/JNEUROSCI.1869-10.2010), suggesting that the effects identified by the authors are not specific to acetyl choline. If the effects of EI are presumed to relate to recurrent pyramidal neurons, these connections could take the form of many different classes of cells (doi.org/10.1038/s41593-020-0685-8)

• Have the authors tested the stability/fit of their model following the addition of the new inhibitory connection? It’s possible that, by adding this gain, the authors have fundamentally altered the fit to LFP data from the original study. This could potentially explain why the authors required different parameter combinations (e.g., Beta = 0.4 in Fig 4B) to recover the original results from Shine et al (2018).

Reviewer #2: The paper by Coronel-Oliveros et al. describes the effect of including local inhibitory circuits and their cholinergic modulation on segregation/integration balance in a modified Jansen-Rit model. These newly added local inhibitory coupling between interneurons is a variation of modelling inhibition-mediated control of the E/I balance which is an interesting approach to modelling neuromodulatory systems.

General comment:

While the methods are profound, the graphs are of a high quality and the language of the manuscript is very understandable, the manuscript would benefit from clearly stating the essential message and sticking to its focus by restricting the number of exploratory analyses. In particular, while the introduction clearly states the interest in integration and segregation as a function of cholinergic action in inhibitory interneurons, also parameters such as Kuramoto order parameter, signal-to-noise-ratio, regularity, static FC, FCD, FCD speed, multistability (not even discussed in the methods), phase synchrony, mean oscillatory frequency and many more have been added. To me, the advantage of so many analyses and a wealth of companyingvery similar figures 2-5, 7 is not clear. Rather

this approach seems like a rather exploratory analysis of all these parameters (that were taken from various computational neuroscience papers) without a clear focus and this approach not only dilutes the mean focus of the paper, makes all these values and their dependence on alpha and beta much harder to interpret. The authors should restrict the parameters to those who answer the initial question of the manuscript and also rewrite the results and discussion section with a clearer focus to make their message more understandable. If more parameters than the original ones (segregation, integration) are chosen, it should be stated in the introduction on how they contribute to answering the original question.

Detailed comments:

The results section includes many interpretations of the analyses (e.g. line 167, 161, 184, 172-183, etc). The authors should move these interpretations to the discussion section to make the results section more concise.

L. 162/203/etc: at times, different values when fixing alpha and beta were chosen. What was the rationale between switching the fixed parameters? It would make sense to stick with the same fixed parameters across the different analyses and explain why these exact values were chosen.

L. 498: what is the advantage to simulating a BOLD signal for these analyses? Why not stick with the EEG signal?

Fig. 3: is along the lines of my first comment. I do not see the main message of this figure and how it contributes to answering the original question

Fig 1 and 4 seem almost redundant to me- in my view, the manuscript would benefit from reducing the number of figures.

L. 209: with a wealth of parameters, the critical boundaries

Discussion:

When reading the discussion, I found it hard to understand the main focus of it- partly it is reads like an enumeration of ideas that are only partly connected. As I said in my first comment, the manuscript would benefit from a more precise and restricted analysis and in a similar fashion from a more focussed discussion focussing on the original question.

L. 298: Please sum up the main results in the first section of the discussion

L. 309: the referenced papers (11,19) do not include any analyses of integration. Which experiments do you mean exactly and what did they really show? So far, I have only seen this relationship in whole-brain models only. Please add experimental papers that were dealing with neuromodulation and segregation/integration.

L. 332: which results back that up? I missed the inverted U across the large amount of figures. Here, I again reiterate to reduce the number of figures according to your main question and focus on the figures with the main message.

L. 402: here two more parameters are introduced that should be removed from the manuscript to sharpen the focus on the original question

Code/Data availability:

The authors provided a github repo with the accompanying code to reproduce the simulations. While I did not run the simulations myself, I found the code to be very well written and understandable and well documented. In addition to the provided codes, it would be good to include the measures of integration and segregation into the code as these are the main variables of interest in the paper.

Reviewer #3: The authors Coronel-Oliveros, Cofré, and Orio, performed a parameter exploration of a whole-brain network model utilizing local dynamics from a neural mass model to describe the effects of neuromodulatory systems and functional segregation and integration in the brain on the source network level and the level of BOLD and EEG. The presented study heavily relies on Shine et al., 2019, and the modeling work Shine et al., 2018. The authors extended the modeling work of Shine et al., 2018 by using a different local dynamic model. The authors considered two neuromodulatory systems, the cholinergic and the noradrenergic system. Both are assumed to act uniformly in the brain at the level of cortical columns. The cholinergic system is also considered to modulate the connectivity via white-matter fiber tracts. Both systems are assumed to act independently and on a slower time scale than the local dynamics. In a first modeling study, the authors systematically varied the level of local connectivity of inhibitory to excitatory interneurons and the level of long-range connectivity and assessed integration and segregation by graph-theoretic measures. In a successive study, the authors performed a similar analysis. They varied the level of long-range connectivity and the variance of firing thresholds for fixed connectivity levels from inhibitory to excitatory interneurons. The main result is that the cholinergic system action on both the long-range connectivity and the inhibition of excitatory interneurons is needed to shift from an unsynchronized regime towards a coherent activity (integrated). The model predicts that the projection of inhibitory to excitatory interneurons is important for controlling the dynamics of a brain area.

I appreciate that kind of modeling work. However, the paper in the present version misses a proper description of observed phenomena associated with neuromodulation through the cholinergic and the noradrenergic system. Effects are often too vaguely described, and it is not clear how they are reflected in brain signals such as EEG and BOLD. I also miss a convincing motivation for the used model. For instance, the term neuromodulation repeatedly appears in the text, but the associated parameters are constants and do not change with time in the model. For the reader, it is important to know how constant model parameters and neuromodulation go together. The authors show the effect of the model parameters on the graph-theoretic measures (efficiency and modularity) and dynamic functional connectivity to assess integration and segregation of functional network states. However, the authors do not show and mention any particular functional networks. I am curious to see the occurring networks and how meaningful they are. I unfortunately cannot recommend the manuscript in its current stage for publishing in Plos-CB.

**Major comments

My reservations concern the model and the description of the neuromodulation systems, most of which can be addressed by improving and elaborating the text's description.

The description of the cholinergic and the noradrenergic systems should be clearer and more consistent throughout the paper. On the one hand, the authors should elaborate more on the physiological aspect - why these systems are so important? On the other hand, the authors should better motivate neuromodulation's modeled action in the local dynamic model.

From the author's description of the biophysical mechanisms, I could imagine other implementations for the action of both systems, for example,

*the cholinergic system :

"lines 56/57, increasing the excitability of pyramidal neurons" could be modeled by lowering firing threshold theta of pyramidal cells and by increasing PSP gain, that is, C_2, C_4, and C at pyramidal cells. Why have the authors decided to scale the input from other brain areas to describe pyramidal cells' increased excitability? In the model, input from other brain areas is linear in the PSP at pyramidal cells. I agree that "lines 70/71: pyramidal neurons become more responsive to stimulus from other distant regions respect to the stimulus of its own cortical column." However, is that equivalent to "increasing the excitability of pyramidal neurons"? Please elaborate.

"Lines 67/68, enhancing the activity and firing rates of dendritic-targeting GABAergic interneurons" should be modeled by beta*x_2(t) at excitatory interneurons as well as pyramidal cells. Why is the presented model beta*x_2(t) affecting excitatory interneurons only and not the inhibitory projections onto pyramidal cells?

*the noradrenergic system :

“Increases responsiveness to input-driven activity respect to spontaneous activity and filters out noise.” In my opinion, that action is better represented in the model by the scaling alpha of the connections between brain areas, also because the "lines 51/52: the effect is more pronounced between distant brain regions." How do the authors relate this noradrenergic effect to a slope change in converting postsynaptic potentials to the firing rate? A flat slope allows for a linear transfer of potential dynamics into rate dynamics. A steep transfer function restricts the dynamic range of the conversion. Therefore, a neural mass is more likely to be saturated: the saturated unexcited or in the saturated excited state. In both states, the neural mass is less sensitive. In general, the slope reflects the variance of the firing thresholds theta within a neural mass. That's why I am curious to know why the authors have decided to alter the slope of potential-to-rate function for all neural masses? Please explain.

The authors should elaborate on the model decision. Most of the cholinergic neurotransmission is known on the level of neurons but the authors used a neural mass model instead of a neuronal model. Neurotransmitters are not directly implemented in neural mass models. The associated neurotransmitters do act on different time scales (see, Shine, 2019). What are the neurotransmitters that potentially drive the constant level changes in the presented model. The authors should answer the question of why the level of neural masses and large-scale brain networks is appropriate for studying neuromodulation and functional integration and segregation in general? For instance, it is unclear how connectivity speed is derived from the human connectomes. Here, I guess, the authors confused transmission delay/time with transmission speed. Please clarify and describe how to derive values with unit 1/seconds from a distance (which distance measure was used?).

It is also unclear why the long-range connectivity speed (delay) affects the characteristic time constant of (dendritic) excitatory postsynaptic potentials at pyramidal cells? The impulse response functions h_E(t) and h_I(t) are properties of local neural masses such as the pyramidal cells. In contrast, long-range connectivity is a network property. The characteristic time constant of postsynaptic potentials differs dependent on the target of synapses on the dendrite. Studies on the single pyramidal cells show that inhibitory interneurons target closer to the soma and excitatory interneurons more distal. Excitatory synapses from more distant areas target more distal and show a distinct characteristic time constant in the postsynaptic potentials. So I agree to the extent that long-range input is integrated with a different time, but I do not see the point of having a different time constant for different lengths of long-range connections. Please elaborate. Time delays tau would read x_{3,ij}(t-tau_{ij}) in the equation system (1).

What are the model assumptions? The modulatory systems do show spatial organization (e.g., https://doi.org/10.1073/pnas.1703601115). Is that an approximation? I suggest adding paragraphs discussing model assumptions, expectations, data descriptions, predictions, and how to test these.

**My specific comments (reading the manuscript from the beginning to the end).

Line 3: What is an optimal behavioral outcome?

Line 5: This statement should be softened. There are also other potential candidates for describing state changes (multistability dynamics, structured flow on manifolds, etc.).

Line 8: The point that "segregation/integration balance is impaired in several neuropsychiatric disorders" should be discussed in more detail in the main text. How is such an impairment of segregation/integration balance reflected in brain signals? What are the relevant disorders?

Line 35: Please provide a reference for the "non-stationarity" of functional connectivity. Are fluctuations at rest non-stationary or non-linear? See, for example, https://dx.doi.org/10.3389%2Ffnins.2020.00493

Line 40: "Neuromodulatory systems provide a biophysical mechanism that enhances the dynamical flexibility." What are these systems? Please provide examples? It appears to be a category of several systems that are capable of modulating neurons. Also, a definition of "dynamical flexibility" is missing.

Line 44: "Indeed, the cholinergic system increments " That is one specific neuromodulatory system. It reads like there is no other. Please summarise for the reader what the "cholinergic system" is and elaborate on the neurophysical role of this system and its elements.

Line 49: What is the "noradrenergic system"? Please elaborate.

Line 50: "input-driven activity" What is the input? Do you mean stimuli like visual stimuli? Do you mean any input that a neural population receives other than its intrinsic "spontaneous activity"? Please clarify.

Line 67: Please clarify the difference between "activity and firing rates"?

Lines 67 to 71: ".. and second, enhancing the activity and firing rates of dendritic-targeting GABAergic interneurons, an effect that promotes intra-columnar inhibition, reducing the local excitatory feedback to pyramidal neurons [23,26,27]. This reads like "reducing the local excitatory feedback to pyramidal neurons" is a necessary reaction of the increased "intra-columnar inhibition." That is not necessarily the case because the pyramidal cells are also affected by intra-columnar inhibition.

Lines 89 to 91: “.. "excitatory gain,” which increases the inter-columnar coupling. This gain mechanism is mediated by the action of the cholinergic system in pyramidal neurons, principally but not exclusively, and increments pyramidal excitability [10, 11, 22]." Is it not the noradrenergic system that acts on a large-scale between brain areas, as mentioned before?

Lines 95 to 99: "Finally, we incorporated a “filter gain,” that increments the pyramidal neurons sigmoid function slope [11]. The noradrenergic system mediates this last gain mechanism; it acts as a filter, decreasing (increasing) the responsivity to weak (strong) stimuli [15,17], boosting signal-to-noise ratio and promoting integration [10]." The described actions are local and equal for all neural masses. Still, the effect is described to be long-range "lines 51 to 52: This effect is more pronounced between distant brain regions, in which structural connectivity is relatively low, promoting functional integration." Please clarify.

Lines 120/121: The time delays are not defined in the Materials and Methods section. If time delays are defined by the distance between brain areas divided by a speed, please clarify and discuss the assumed speed (is it a spatially invariant constant). What distance measure was applied (Euclidean as a lower bound proxy or mean streamline length?). Please elaborate.

Page 4, Fig. 1. "The cholinergic system has a multiplicative effect on the sigmoid function. α amplifies pyramidal neurons' response to other columns’ input" What do the authors mean exactly with multiplicative effect on the sigmoid function? Please explain in the main text? Please clarify "response"? Do the authors refer to postsynaptic potentials or firing rate?

Lines 127 to 139: The authors should emphasize the model parameters and that these are constant levels for each simulation. I understand the presented model in that way that there is no neuromodulation. The model parameters alpha, beta, and r_0 are constants and are no functions of time as readers might expect from "neuromodulation." The authors should highlight time scales, separate them, and why and under which circumstances the systematic exploration of constant parameters matters. Because this is so important for the modeling work, this should be mentioned and discussed at several stages in the manuscript.

Lines 242 to 244: There is a difference between noise and chaos in the model. A noise process drives the model with predefined moments. Chaos can occur due to the model's complexity and the coupling in the network (see https://doi.org/10.1371/journal.pcbi.1002298 and https://doi.org/10.1016/j.neuroimage.2016.02.015). Although the applied measure does not distinguish between noise and chaos, the system's ability to show deterministic chaos should be discussed. Whereas the noise process represents an additional dimension and something 'unknown' extrinsic, the deterministic chaos is intrinsic and produced and maintained by the system itself.

Lines 259 to 297: The authors have to define the term criticality? Is it a statistical term, or does it correspond to deterministic mechanisms such as bifurcations that occur in the local dynamic model? Please elaborate.

Line 299 to 306: The authors refer to experimental findings based on the action of nicotinic acetylcholine (20,23,27) and somatostatin receptors (26) on spiking single neurons. How do these electrophysiological findings translate into the hypothesis that "cholinergic neuromodulation of the inhibitory interneurons (that suppresses the local 300 excitatory feedback to pyramidal neurons) facilitates functional integration?" Moreover, how can the utilized mesoscopic - large-scale level of neural masses and long-range connectivity help test the hypothesis. Why do the authors use forward models for EEG and BOLD? Do EEG and BOLD data exist supporting the hypothesis linked to the electrophysiological findings?

Line 307 to 310: The references 11 and 19 are reviews, so I wonder which extend the presented model can explain the, in refs 11 and 19, discussed experimental data. Does the model explain more than the already described and modeled inverted U-shape (10,18)?

Lines 317/18: Again, the modes as presented do not include time delays. In the model, the transmission times of long-range connections determine the local characteristic time scale of the excitatory postsynaptic potentials at the receiving pyramidal neurons. Here, the authors should give motivation for that implementation in the model. To me, it does not sound biophysically plausible.

Lines 449: How important are the mean and variance of the noise process for the results? What is the effect of noise?

Lines 465 to 468: Do the authors really mean speed here? The physical SI-unit for speed is meter/second. So I wonder, how is the speed (m/s) derived from the connectome? Usually, the distance is decided by a constant speed to approximate transmission delays. However, to include transmission times as characteristic time constants in the ODEs is also not correct as these are two different things. The characteristic times in the ODEs, in fact, h_{E, I}(t) represent local properties of the postsynaptic responses to synaptic input and should not vary for different incoming connections. Please elaborate. These points need to be clarified. What is the interpretation of the distances between brain areas here? Are we talking about Euclidean distances or mean streamline lengths between brain regions?

Pages 13- The equations in the Materials and Methods section should be consistently numbered. In the presented manuscript are 17 equations, but only three equations have numbers.

Page 13, equation (1):

There are four sets of 2nd-order ODEs. For better understanding, it is worth describing their function.

I. The first two equations are for the excitatory projections of pyramidal cells onto both interneurons.

II. The second pair of equations is for the excitatory projections of excitatory interneurons onto pyramidal cells. Here we see, that the external input is assumed to be excitatory and share the characteristic time constant of intra-areal excitatory projections at the pyramidal cells.

III. The third pair of equations is for the inhibitory projections of inhibitory interneurons onto pyramidal cells.

IV. The fourth pair is very similar to the I pair of equations only differs in the scales and is meant to explain the excitatory long-range projections of pyramidal cells in distant areas onto target pyramidal cells. Here the time constants depend on the length of incoming connections, for which an explanation is missing. Please elaborate. The notation of the incoming activity to y_0 and y_3 are identical (within the sigmoids). This becomes circular with inserting z_i(t) (the unnumbered equation below). I guess, the input to y_3 should read ( C_2 x_{1,j} - C_4 x_{2,j} + C alpha z_j ) because the sigmoid looks backward into the source. That's why the equation for the average input should also be adapted to, for instance, z_{a,b} = sum^{n}_{b=1} M_{ab} x_{3,ab}, where a,b are indices of brain areas and b is the source whereas a is the target index.

The authors should elaborate on the fact that the connectivity weights are normalized individually per receiving brain area. Why is that? The equation for the average input can be simplified M_{ab} = Mij/sum_{j} (M{ij), where Mij is the weight as presented in the manuscript. An overall normalization by a scalar uniformly applied to all entries, for instance, the maximum in-strength sounds more plausible. By using an input wise normalization, the relative weights between receiving brain areas are lost.

The equations show that the inhibitory interneurons have two targets: the pyramidal cells and the excitatory interneurons. The authors motivated the scaling of inhibitory activity as neuromodulation. Why is it then that only the inhibitory postsynaptic potential at the excitatory interneurons are scales but not the inhibitory postsynaptic potential at the pyramidal cells? However, the source of activity is identical? This modeling choice must be better motivated.

Line 473: the maximum firing rate zeta should be in 1/s - it's a rate, not a frequency.

Lines 475 to 477: Why is only the firing rate of pyramidal cells used to calculate BOLD? Mainly, the pyramidal cells' postsynaptic potential is reflected in M/EEG because of the number of pyramidal cells and their arrangement. However, that does not apply to BOLD. Here all neural masses contribute.

**Have all data underlying the figures and results presented in the manuscript been provided?**

Reviewer #1: Yes

Reviewer #2: Yes

Reviewer #3: Yes

PLOS authors have the option to publish the peer review history of their article (what does this mean?). If published, this will include your full peer review and any attached files.

Reviewer #1: No

Reviewer #2: No

Reviewer #3: **Yes: **Andreas Spiegler
---

## [Decision Letter · Decision Letter 1]

19 Jan 2021

Dear Mr. Orio,

Thank you very much for submitting your manuscript "Cholinergic neuromodulation of inhibitory interneurons facilitates functional integration in whole-brain models" for consideration at PLOS Computational Biology. As with all papers reviewed by the journal, your manuscript was reviewed by members of the editorial board and by several independent reviewers. The reviewers appreciated the attention to an important topic. Based on the reviews, we are likely to accept this manuscript for publication, providing that you modify the manuscript according to the review recommendations.

Sincerely,

Daniele Marinazzo

Deputy Editor

PLOS Computational Biology

[LINK]

Reviewer's Responses to Questions

**Comments to the Authors:**

Reviewer #1: The authors have adequately addressed my concerns.

Reviewer #2: Thank you for inviting me again to review this manuscript on neuromodulatory systems and their relationship with segregation and integration. I read the manuscript in great detail and I was very pleased with the changes, esp. with the new version of the discussion and I enjoyed re-reading the manuscript, this time with a better understanding of the different hypotheses and arguments raised by the authors. My major concern, the lack of scope, has been addressed throughout the manuscript, so that I was able to easily follow the leitmotif provided by the authors. I am very pleased that my concerns and suggestions have been addressed and overall I feel that the manuscript has improved substantially after the revision. I only have a few minor remarks (the line count refers to the version that highlights the changes):

Introduction:

I found the introduction, in comparison to the overall length of the manuscript, a little bit lengthy. Making it more concise could improve the readability of the article.

l. 119: Which experimental findings are being referred to? Later in the manuscript the authors rather rague that they want to provide two time scales- I find this argument more convicing

Fig.1: It would make sense to the modification of the model more visible (e.g. using a color), to help the reader to directly see the changes

Fig. 3: It would be great to associate the red circles to the letters within the picture (e.g. B) to faster see which circle relates to which graph

l.428- As this analysis is based on a simulation, I would rather not talk about "experiment"

l.598- It might be more advantageous for the authors' argument to name diseases with a known cholinergic deficit (e.g. Alzheimer's or Parkinson's disease dementia) and noradrenergic deficit (e.g. ADHD)

Discussion:

I especially enjoyed reading the new version of the discussion

**Have all data underlying the figures and results presented in the manuscript been provided?**

Reviewer #1: None

Reviewer #2: **No: **

PLOS authors have the option to publish the peer review history of their article (what does this mean?). If published, this will include your full peer review and any attached files.

Reviewer #1: No

Reviewer #2: No
---

## [Editor Report · Decision Letter 2]

25 Jan 2021

Dear Mr. Orio,

We are pleased to inform you that your manuscript 'Cholinergic neuromodulation of inhibitory interneurons facilitates functional integration in whole-brain models' has been provisionally accepted for publication in PLOS Computational Biology.

Best regards,

Daniele Marinazzo

Deputy Editor

PLOS Computational Biology

Daniele Marinazzo

Deputy Editor

PLOS Computational Biology

---

## [Editor Report · Acceptance letter]

11 Feb 2021

PCOMPBIOL-D-20-01747R2 

Cholinergic neuromodulation of inhibitory interneurons facilitates functional integration in whole-brain models

Dear Dr Orio,

I am pleased to inform you that your manuscript has been formally accepted for publication in PLOS Computational Biology. Your manuscript is now with our production department and you will be notified of the publication date in due course.

With kind regards,

Alice Ellingham
